# Unveiling unique microbial nitrogen cycling and nitrification driver in coastal Antarctica

Ping Han [1,2,3], Xiufeng Tang[1], Hanna Koch[4,5], Xiyang Dong [6,7,8], Lijun Hou [2,3], Danhe Wang[1], Qian Zhao[1], Zhe Li[1], Min Liu [1,3] ✉, Sebastian Lücker [4] & Guitao Shi [1,2] ✉

Largely removed from anthropogenic delivery of nitrogen (N), Antarctica has notably low levels of nitrogen. Though our understanding of biological sources of ammonia have been elucidated, the microbial drivers of nitrate ($NO_3^-$) cycling in coastal Antarctica remains poorly understood. Here, we explore microbial N cycling in coastal Antarctica, unraveling the biological origin of $NO_3^-$ via oxygen isotopes in soil and lake sediment, and through the reconstruction of 1968 metagenome-assembled genomes from 29 microbial phyla. Our analysis reveals the metabolic potential for microbial $N_2$ fixation, nitrification, and denitrification, but not for anaerobic ammonium oxidation, signifying a unique microbial N-cycling dynamic. We identify the predominance of complete ammonia oxidizing (comammox) *Nitrospira*, capable of performing the entire nitrification process. Their adaptive strategies to the Antarctic environment likely include synthesis of trehalose for cold stress, high substrate affinity for resource utilization, and alternate metabolic pathways for nutrient-scarce conditions. We confirm the significant role of comammox *Nitrospira* in the autotrophic, nitrification process via [13]C-DNA-based stable isotope probing. This research highlights the crucial contribution of nitrification to the N budget in coastal Antarctica, identifying comammox *Nitrospira* clade B as a nitrification driver.

Antarctica, a region largely insulated from anthropogenic nitrogen (N) deposition, exhibits notably low N concentrations[1]. These diminished N levels play an integral role in sustaining ecological equilibrium in the continent's barren, ice-free terrains[2,3]. Consequently, elucidating the intricacies of microbial N-cycling pathways in Antarctic ecosystems—including diazotrophy ($N_2$ fixation), nitrification, anaerobic ammonium oxidation (anammox), and denitrification—is paramount[4]. Diazotrophic Cyanobacteria are the primary producers of biologically accessible ammonium ($NH_4^+$) in these environments[5]. Alongside $NH_4^+$, nitrate ($NO_3^-$) constitutes a significant inorganic N pool in polar biomes[6,7]. The synthesis of $NO_3^-$ in Antarctic ecosystems can be attributed to both abiotic atmospheric deposition, particularly within mineral-rich Antarctic soils[8], and biotic nitrification processes. Still, the discrete contributions of these sources to the coastal Antarctic $NO_3^-$ reservoir remain inadequately characterized, necessitating further investigation.

[1]Key Laboratory of Geographic Information Science (Ministry of Education), School of Geographic Sciences, East China Normal University, 500 Dongchuan Road, Shanghai 200241, China. [2]State Key Laboratory of Estuarine and Coastal Research, East China Normal University, 500 Dongchuan Road, Shanghai 200241, China. [3]Institute of Eco-Chongming (IEC), East China Normal University, 3663 North Zhongshan Road, Shanghai 200062, China. [4]Department of Microbiology, RIBES, Radboud University, Heyendaalseweg 135, 6525 AJ Nijmegen, the Netherlands. [5]Center for Health & Bioresources, Bioresources Unit, AIT Austrian Institute of Technology GmbH, A-3430 Tulln, Austria. [6]Key Laboratory of Marine Genetic Resources, Third Institute of Oceanography, Ministry of Natural Resources, Xiamen 361005, China. [7]State Key Laboratory Breeding Base of Marine Genetic Resources, Xiamen 361005, China. [8]Fujian Key Laboratory of Marine Genetic Resources, Xiamen 361005, China. ✉e-mail: mliu@geo.ecnu.edu.cn; gtshi@geo.ecnu.edu.cn

Nitrification plays a pivotal role in the global biogeochemical N cycle[9] and contributes significantly to the emissions of nitrous oxide ($N_2O$)[10], a potent greenhouse gas. The process of nitrification encompasses the oxidation of ammonia ($NH_3$) to $NO_3^-$ via the intermediate nitrite ($NO_2^-$). Traditionally, these two metabolic steps have been attributed to distinct microbial guilds: ammonia-oxidizing bacteria (AOB)[11] and archaea (AOA)[12], responsible for converting $NH_3$ to $NO_2^-$, and nitrite-oxidizing bacteria (NOB)[13], which facilitate the subsequent oxidation of $NO_2^-$ to $NO_3^-$. Beyond these canonical nitrifiers, the recently identified complete ammonia oxidizers (comammox) within the genus *Nitrospira*, specifically lineage II, perform full nitrification on its own[14,15]. These organisms are increasingly recognized for their significant role in nitrification across diverse ecosystems, including both engineered systems[16] and natural environments[17]. The contribution of comammox bacteria to nitrification and subsequent $NO_3^-$ production in Antarctic ecosystems, however, remains an enigma.

Phylogenetically, comammox *Nitrospira* cluster into clades A and B[14], distinguished by their respective ammonia monooxygenase genes (*amoA*). Physiological assessments of clade A comammox *Nitrospira* have revealed an exceptionally high $NH_3$ affinity, suggestive of an oligotrophic niche adaptation[18,19]. In stark contrast, clade B comammox *Nitrospira*, despite their widespread distribution in various habitats, including those characterized by low temperatures such as the Tibetan Plateau[20,21] and Arctic permafrost[22], have yet to be cultured for physiological scrutiny.

This study aims to unravel the complexities of the microbial N cycle in the ice-free regions of East Antarctica, identifying the primary sources of $NO_3^-$ and examining the potential roles of microorganisms in the nitrification process. Our research is strategically centered on the Larsemann Hills (LH)[23], a vast ice-free rocky landscape in East Antarctica, dotted with over a hundred oligotrophic lakes shrouded in ice. Through a comprehensive approach, we discover that (i) the origin of $NO_3^-$ is primarily the biological nitrification process; (ii) the microbial N cycle is distinct, encompassing most microbial N-cycling processes except for the anammox pathway; and (iii) the comammox *Nitrospira* clade B serves as an abundant and active driver of nitrification, possessing various survival strategies against the cold and oligotrophic conditions of the coastal Antarctic environment.

## Results and discussion
### Isotopic signatures indicate the biological origin of nitrate
Depending on how $NO_3^-$ is produced, the composition of its oxygen isotopes differs, allowing differentiation between abiotic and biotic sources[24]. For biologically produced $NO_3^-$, one oxygen atom (O) is expected from atmospheric oxygen ($O_2$) and two from the surrounding water ($H_2O$)[24]. Biologically produced $NO_3^-$ would have a $\delta^{18}O$ of ~0.6‰, since $\delta^{18}O$ of atmospheric $O_2$ is 23.9‰[25] and the measured $\delta^{18}O$ of Larsemann Hills (LH) lake water is $-12.7 \pm 1.5$‰ (Supplementary Table 1). In addition, while biologically produced $NO_3^-$ has a $\Delta^{17}O$ of 0‰, which is identical to that of $O_2$ and $H_2O$, atmospheric $NO_3^-$ is usually characterized by high oxygen isotopic ratios[26,27] and can reach $\Delta^{17}O \geq 35$‰ in the atmosphere in Antarctica[26]. Atmospheric $NO_3^-$ in Antarctica is mainly produced by the reaction of nitrogen oxides ($NO_x$), ozone ($O_3$), and hydroxyl radicals (OH·)[27].

The mean $\delta^{18}O$ of $NO_3^-$ in sediments from LH lake is 4.6‰. The annual mean $\delta^{18}O$ of $NO_3^-$ in snow and the atmosphere at the coastal Zhongshan station, situated in LH, is approximately 92‰[28]. From these, the contributions of atmospheric deposition and biological production to sedimentary $NO_3^-$ pools can be quantified by isotope mass balancing, indicating that the nitrification process constitutes 95% of the $NO_3^-$ in sediments. In addition, limited data of sediment $NO_3^-$ $\Delta^{17}O$ produced a mean of ~1.3‰, which strengthens the observed >90% contribution of nitrification considering the very high $\Delta^{17}O$ of atmospheric $NO_3^-$ at Zhongshan station (~32‰)[28]. Similarly, the

isotope mass balance of $\delta^{18}O$ and $\Delta^{17}O$ suggested ~96% of soil $NO_3^-$ in the two study areas is from nitrification. Thus, we demonstrated here that sediment and soil $NO_3^-$ predominantly originated from biological production in and nearby the investigated lakes.

### Unique microbial nitrogen cycle in coastal Antarctica
The primary production of $NO_3^-$ is largely attributed to the microbial nitrification process, rather than atmospheric precipitation. Consequently, the origin of $NH_4^+$, the substrate for nitrification, can be traced back to biological $N_2$ fixation, as well as mineralization. The qPCR-based quantification of functional N-cycling genes (Fig. 1) revealed a comparable amount of $N_2$ fixation (*nifH*, encoding nitrogenase) to nitrification genes (*amoA* and *nxrB*, encoding ammonia monooxygenase subunit A and nitrite oxidoreductase subunit B, respectively), with quantities ranging from $10^2$ to $10^4$ copies ng$^{-1}$ DNA. The relatively low quantity of the gene encoding hydroxylamine dehydrogenase (*hao*) (<10 copies ng$^{-1}$ DNA) can be attributed to the low coverage of the applied primers (targeting AOB rather than comammox *Nitrospira*, Supplementary Table 2) and the absence of the bacterial *hao* gene in the genomes of AOA[29], which are of notable abundance in the studied samples.

In addition to $N_2$ fixation and nitrification processes, we also observed the significant metabolic potential of denitrifiers, as indicated by the high abundances (>$10^3$ copies ng$^{-1}$ DNA) of genes encoding the sequential reduction of $NO_3^-$ (*nar*, *nap*), $NO_2^-$ (*nir*), nitric oxide (NO, *nor*), and $N_2O$ (*nos*) (Fig. 1). The relatively higher abundances (>$10^4$ copies ng$^{-1}$ DNA) of *nirS* and *nirK* genes (Fig. 1) can likely be attributed to their presence not only in denitrifiers but also in nitrifiers. Interestingly, we found substantial quantities ($10^2$ to $10^4$ copies ng$^{-1}$ DNA) of the functional gene for dissimilatory nitrate reduction to ammonium (DNRA, *nrfA*) and assimilatory nitrite reduction (ANR, *nasA*) (Fig. 1). The potential for DNRA and ANR is widespread among phylogenetically diverse microorganisms and these processes ensure the retention of bioavailable inorganic N in an ecosystem.

Strikingly, the *hzo* gene (encoding hydrazine oxidoreductase), a biomarker for the anammox process, was not detected in any of the tested sediment and soil samples. Anammox bacteria catalyze the anaerobic oxidation of $NH_4^+$ using $NO_2^-$ as an electron acceptor, producing $N_2$ as the final product. These bacteria were first identified in wastewater treatment systems[30] and were subsequently discovered in various environments, including marine[31], coastal[32], terrestrial[33], and engineering systems[34]. The absence of anammox functional markers in coastal Antarctica suggests unique microbial N-cycling properties in this remote region.

### Diverse microbiomes and the microbial N-cycling processes
From an extensive dataset exceeding 450 gigabases of sequencing data, we managed to reconstruct a dereplicated collection of 724 high-quality and 1244 medium-quality[35] metagenome-assembled genomes (MAGs) (Supplementary Data 1). The recovered genomes encompass 29 distinct phyla (Supplementary Data 1 and 2), marking, to the best of our knowledge, the most comprehensive inventory of Antarctica coastal soil and sediment genomes to date. On average, we obtained around 60 MAGs per sample site studied (Fig. 2a). The most abundant phyla included Actinobacteriota, Pseudomonadota, Bacteroidota, Chloroflexota, Verrucomicrobiota, Acidobacteriota, Patescibacteria, Planctomycetota, and Gemmatimonadota (Fig. 2a and Supplementary Data 1), aligning with findings from other Antarctic surveys[5,36,37]. Cyanobacteria also featured among the top ten most abundant MAGs (Supplementary Data 1), a deviation from the Mackay Glacier Region, where they were largely absent in most soil samples[37]. We identified only five archaeal MAGs within the Thermoproteota phylum, all of which were further classified as ammonia-oxidizing archaea (AOA) within Group I.1b.

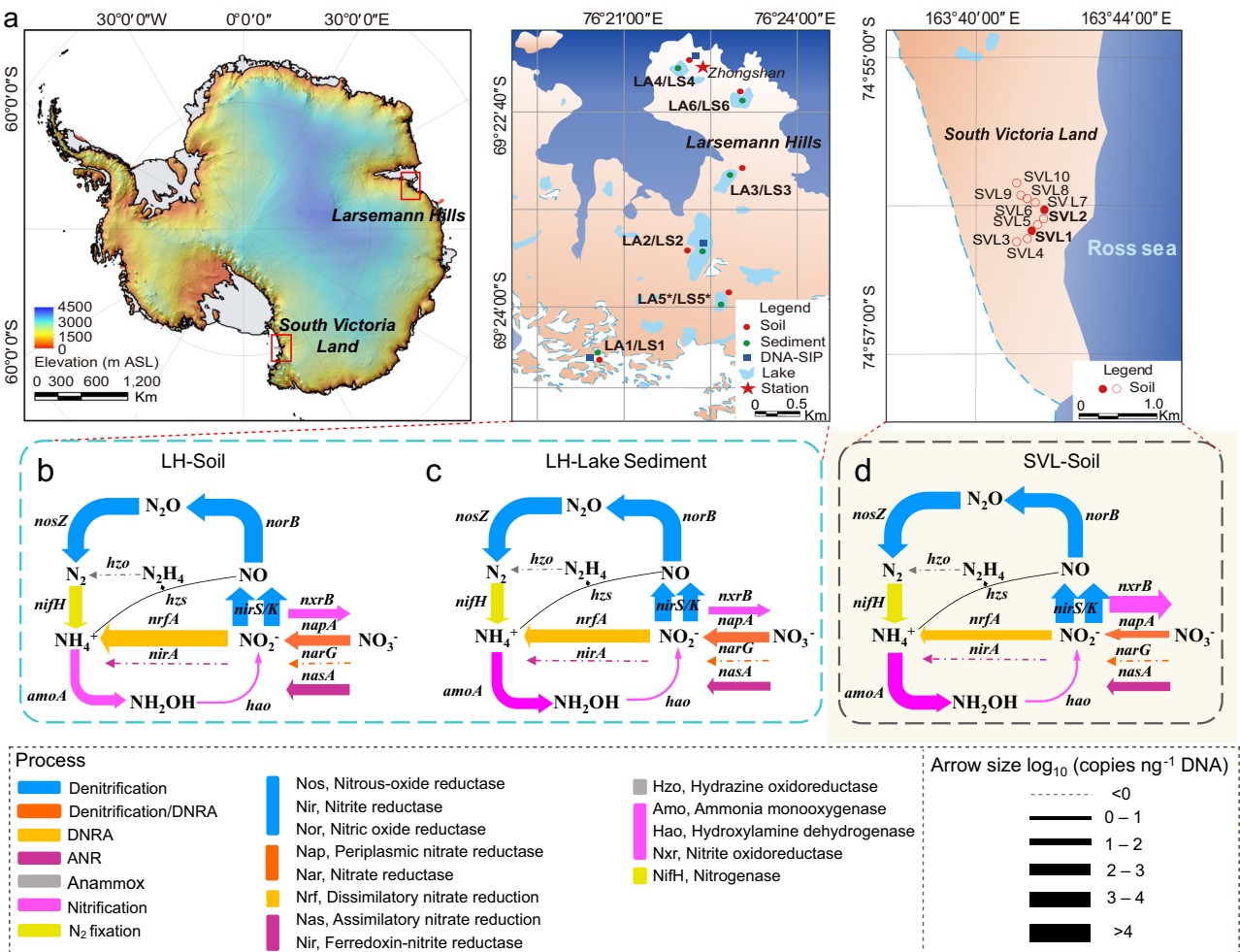

**Fig. 1 | Overview of sampling sites, sample types, and quantities of functional nitrogen-cycling genes in coastal Antarctica. a** A photograph illustrating the location of the Larsemann Hills (LH) and South Victoria Land (SVL) (left) in Antarctica, locations of studied surface soil and lake sediments in LH (center), and locations of surface soil samples in SVL (right). Red and green circles represent soil and sediment samples, respectively. Solid circles denote samples that have undergone metagenomic analysis, while empty circles denote samples that have been subjected solely to quantitative PCR analysis. The asterisk indicates samples that were used exclusively for metagenomic analysis and did not undergo comprehensive N-cycling functional gene screening. **b–d** Abundance of functional microbial N-cycling genes in the LH soils (**b**), LH lake sediments (**c**), and SVL soils (**d**). The studied functional N-cycling genes encompass $N_2$ fixation (*nifH*), nitrification (*amoA*, *hao*, and *nxrB*), denitrification (*napA*, *narG*, *nirS/K*, *norB*, and *nosZ*); dissimilatory nitrate reduction to ammonium (DNRA; *nrfA*), assimilatory nitrite reduction (ANR; *nasA* and *nirA*), and anaerobic ammonium oxidation (anammox; *hzo*). The relative abundance of the associated functional genes is indicated by the line thickness. Source data are provided as a Source Data file.

To comprehend the metabolic strategies that sustain the abundant bacterial life in these extremely nutrient-poor sediments and soils, we examined the distribution and affiliation of 52 marker genes conserved across different energy conservation and carbon acquisition pathways[37] in the MAGs we retrieved. As predicted, genes for aerobic respiration were encoded by nearly all community members (Fig. 2b and Supplementary Fig. 1). Consistent with observations from the Mackay Glacier Region[37], a significant number of the MAGs appear to fix carbon via the Calvin-Benson-Bassham (CBB) cycle (212 MAGs) or the 3-hydroxypropionate cycle (345 MAGs) (Fig. 2b). These mechanisms facilitate the generation of biomass through means that are either separate from or complementary to photoautotrophy, a role predominantly carried out by Cyanobacteria (Supplementary Fig. 1). Genomic analysis revealed that the most abundant and widespread community members encoded trace gas oxidation genes. In a pattern similar to previous discoveries[37,38], carbon monoxide (CO) dehydrogenases (CoxL) were exclusive to Actinobacteriota and Chloroflexota (Fig. 2b). Interestingly, uptake [NiFe]hydrogenases were encoded by MAGs from Acidobacteriota, Chloroflexota, and Verrucomicrobiota (Fig. 2b), which aligns with observations from temperate soil where Acidobacteriota are known to be active atmospheric $H_2$ consumers[39], albeit with slight differences.

In the realm of N metabolism, a comprehensive array of functional genes was identified within the MAGs we analyzed (Fig. 2b). $N_2$ fixation appears to be predominantly carried out by taxa within Acidobacteriota, Cyanobacteria, Desulfobacterota, Myxococcota, and Verrucomicrobiota (Fig. 2b). Consistent with the quantitative analyses, the *hzs* gene, indicative of the anammox process, was not present in the retrieved MAGs. While a total of 96 Planctomycetota MAGs were obtained, subsequent verification through taxonomic classification confirmed that none of these MAGs belong to the anammox family Brocadiaceae (Fig. 2b and Supplementary Data 2). The absence of anammox process may be attributed to the optimal temperature range of 12–17 °C, which is characteristic of anammox process found in similarly cold Arctic fjord sediments[40], suggesting a limited tolerance to low temperatures. This assertion is further reinforced by the absence of anammox bacteria in the McMurdo Dry Valleys (an Antarctic desert)[41] and in Arctic regions[42] as well. The pervasive distribution of denitrification genes across nearly all phyla underscores the robust and widespread microbial denitrification activity in Antarctica,

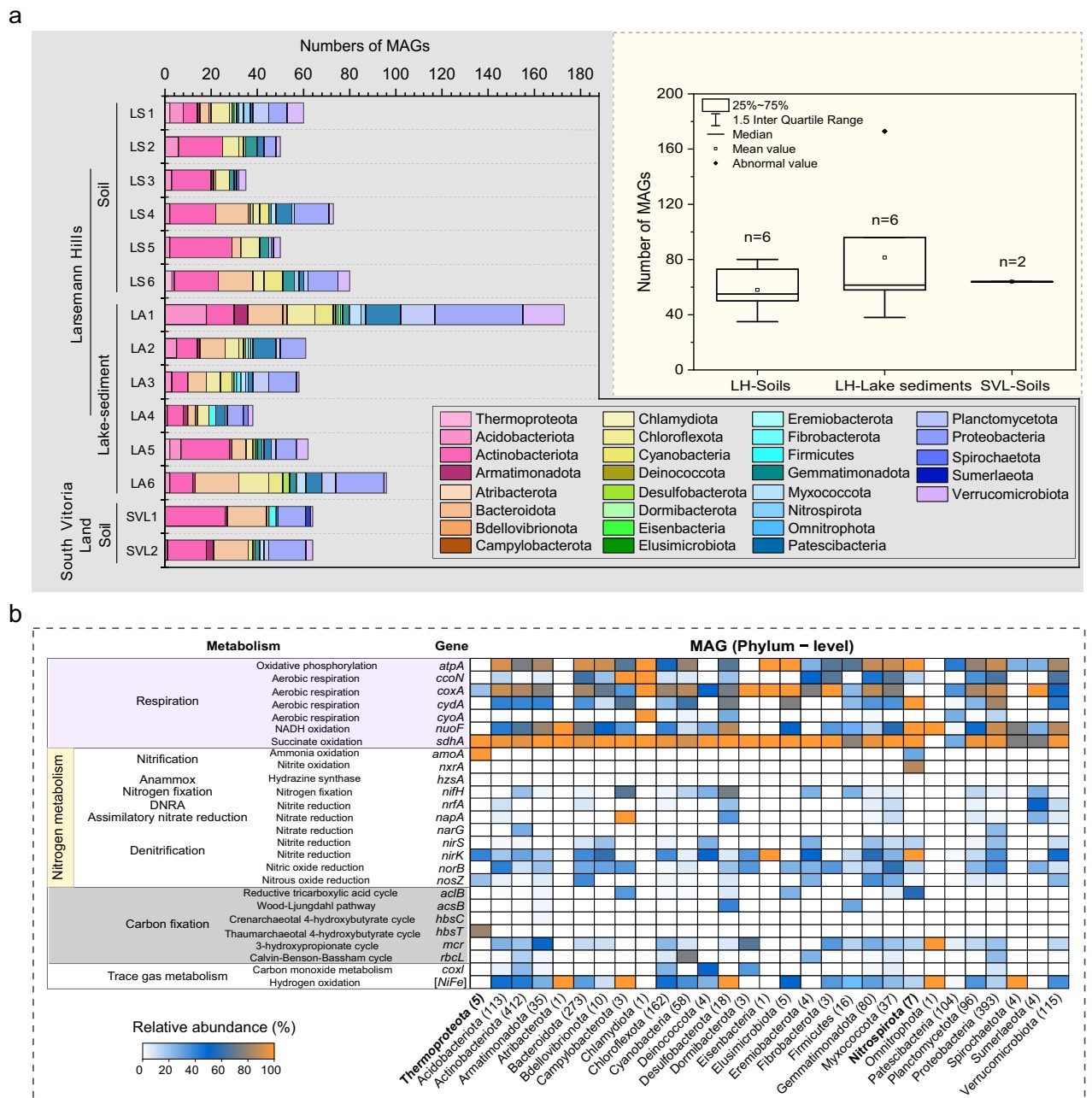

**Fig. 2 | Community composition and relative abundance of selected functional genes derived from metagenomic analyses of various sediments and soils from coastal Antarctica. a** Overview of the microbial community composition derived from metagenome-assembled genomes (MAGs) extracted from six soil and six sediment samples in the Larsemann Hills (LH), as well as two soil samples from South Victoria Land (SVL). The statistical box in the top right corner presents the numerical distribution of the MAGs retrieved from the studied samples (Co-

assemblies are not included in this statistic). **b** Distribution of selected key functional genes in the obtained MAGs, including those involved in respiration, N-cycling, carbon fixation, and trace gas metabolism. The adjacent heatmap displays the distribution of these genes across a set of 1968 MAGs, comprising 29 phyla. Summary and statistics of the presence/absence of key metabolic marker genes in the obtained MAGs are provided in Supplementary Data 1 and 2. Source data are provided as a Source Data file.

corroborating previous research that highlights the ubiquity of cold-adapted denitrifiers across diverse Antarctic ecosystems[43]. Regarding the nitrification process, the marker genes-*amoA* (associated with ammonia oxidation) and *nxrB* (linked to nitrite oxidation) were found exclusively in the Thermoproteota and Nitrospirota phyla (Fig. 2b). Although Antarctic soil nitrification was previously reported back in 1997[44], attributed to AOB genera *Nitrosospira* and *Nitrosomonas*, as well as AOA within group I.1b[45–48], no investigation has yet been conducted into the presence and function of the recently discovered comammox bacteria within the *Nitrospira* genus[14,15]. Here, we present data showing the prevalence of the comammox *Nitrospira amoA* gene

in Antarctica lake sediments and soils, strongly suggesting the activity of comammox *Nitrospira* in this region.

## Abundant nitrification drivers in coastal Antarctica
The quantitative evaluation of nitrification-related functional genes, including *amoA* and *nxrB* (Supplementary Fig. 2), established that AOA and comammox bacteria were the predominant nitrifiers in the analyzed soils and sediments (Supplementary Fig. 3). Our metagenomic analysis corroborates these findings, having identified only AOA and *Nitrospira* MAGs among the various nitrifying groups (Fig. 2b), suggesting they outnumber AOB. Furthermore, a more detailed analysis of

the abundance and community structure of all identified nitrifying groups was conducted using amplicon *amoA* and *nxrB* sequencing. Subsequent phylogenetic assessment revealed an unexpected dominance of clade B comammox *Nitrospira* in this environment (Supplementary Table 5 and Supplementary Fig. 8). This was surprising given the successful extraction of diversity information from AOA (Supplementary Fig. 6) and AOB (Supplementary Fig. 7) in the samples investigated (Supplementary Results and Discussion).

Among the dereplicated MAGs, five were classified as Group I.1b-AOA within the genera *Nitrosocosmicus* and *Nitrospharea* of the Thermoproteota phylum. This categorization stemmed from GTDB taxonomic classification, phylogenomic assessments, and average nucleotide identity (ANI) analyses (Supplementary Data 2 and Supplementary Fig. 4). These findings are in agreement with the data obtained from *amoA* amplicon sequencing, as discussed in the Supplementary Results and Discussion section. Together with their previous detection in Arctic soils[49,50], this suggests a capacity of *Nitrosocosmicus*-like AOA to thrive in cold and nutrient-deficient conditions.

Of the seven *Nitrospirota* MAGs analyzed, four were verified as comammox *Nitrospira*, and the remaining three belonged to the exclusively nitrite-oxidizing lineages II and IV of *Nitrospira*. This classification was substantiated by phylogenomic and average nucleotide identity (ANI) analyses (Fig. 3a, b, Supplementary Table 4, and Supplementary Fig. 5), and these results concur with the phylogenetic assessment based on the *amoA* and *nxrB* gene of *Nitrospira* (Supplementary Figs. 8, 9). The notably high abundance of these *Nitrospira* MAGs in the soil and sediment metagenomes studied (reaching up to 0.173%, Supplementary Table 3) further emphasizes their significant roles in nitrification within these ecosystems.

Two high-quality comammox MAGs, *Nitrospira* sp. La1-X1 and *Nitrospira* sp. La3-X1, have genome sizes of 4.26 Mb and 3.78 Mb, respectively (Supplementary Table 3). The *amoA* gene sequences from La1-X1 and La3-X1 align closely with the dominant *amoA* OTUs and with previously published sequences from clade B comammox (Fig. 4c and Supplementary Fig. 8). Consistent with the *amoA*-based phylogeny, phylogenomic analysis also demonstrates a distinct clustering of these comammox MAGs within clade B (Fig. 3a). For La1-X1, the ANI was highest with MAG *Nitrospira* sp. palsa1310, which was identified from Arctic permafrost soil[22]. The ANI between these genomes is 96.67% (Fig. 3b), surpassing the species threshold of 95%[51], suggesting that these genomes represent different strains of the same *Nitrospira* species, potentially adapted to cold environments. In contrast, La3-X1 is grouped within a clade B subset that includes MAGs from drinking water treatment systems and glacier surface soil[52], but it did not share high ANI values with any other clade B genomes (≤85%, Fig. 3b). This suggests that La3-X1 represents a unique so far undetected group within clade B.

## Metabolic potential and surviving strategy

The clade B comammox MAGs *Nitrospira* sp. La1-X1 and *Nitrospira* La3-X1 harbor the complete genetic machinery for $NH_3$ and $NO_2^-$ oxidation, the respiratory chain, and the reduced tricarboxylic acid (rTCA) cycle, which is the conserved $CO_2$ fixation pathway in *Nitrospira* (Fig. 3c, d). These core metabolic features are highly conserved in comammox *Nitrospira*, as reported in previous studies[53,54]. Similar to other *Nitrospira* genomes[52], La1-X1 and La3-X1 do not encode nitric oxide reductase (NOR), which is crucial for enzymatic $N_2O$ production[55]. They do carry genes for urea transport and hydrolysis by the urease, indicating the use of urea as an alternative ammonia source, as shown for other comammox *Nitrospira*[14,15,56]. In addition, like some *Nitrospira* that have the confirmed ability to use hydrogen and formate as alternative energy sources[57,58], the comammox MAGs La1-X1 and La3-X1 contain genes for formate and hydrogen oxidation. The formate dehydrogenases of comammox are similar to their nitrite-oxidizing counterparts. In contrast, comammox bacteria possess a 3b-type [NiFe] hydrogenase, which

is rarely identified in canonical *Nitrospira* and its physiological role remains unclear[54,59]. While the capacity for formate oxidation is widely distributed in clade B, the 3b-type [NiFe], hydrogenase has been mainly identified in clade A comammox species[54] (Fig. 3d). However, the presence of this hydrogenase type in a few clade B genomes[52,54] including La3-X1 challenges the clade specificity of this feature.

In addition to the canonical $F_1F_0$ $H^+$-driven ATPase, La1-X1 and La3-X1 encode a potentially $Na^+$-pumping $F_1F_0$ ATPase, previously detected in the haloalkalitolerant nitrite-oxidizing *Ca.* Nitrospira alkalitolerans[60], marine nitrite-oxidizing *Ca.* Nitronereus thalassa[61] and clade A comammox *Ca.* Nitrospira kreftii[19]. Similar to these genomes, La3-X1 also possesses a $Na^+$-translocating NADH:ubiquinone oxidoreductase (NQR), a feature missing in La1-X1 (Fig. 3c). These $Na^+$-pumping enzyme complexes might represent an adaption of La3-X1 to saline or haloalkine conditions similar to other *Nitrospira*.

The disaccharide trehalose is one of several solutes known to protect bacteria against cold stress, and may also protect against other harmful environmental conditions, such as osmotic stress[62]. La1-X1 possesses a trehalose-6-phosphate synthase (OtsA) and the corresponding phosphatase (OtsB) to potentially produce the non-reducing disaccharide in two steps from UDP-glucose, as shown for *E. coli*[63]. In contrast, La3-X1 encodes several other trehalose synthesis enzymes in one gene cluster, indicating that this species might convert NDP-glucose (trehalose synthase) and maltose (trehalose synthase/amylase) to trehalose. However, all these trehalose synthesis pathways have also been identified in *Nitrospira* from mesophilic environments and might thus not be a distinguishing feature for cold adaptation. In addition to trehalose synthesis, La1-X1 and La3-X1 possess other features for coping with different stresses. Whereas a gene cluster encoding a nitrile hydratase has been detected in La1-X1 and La3-X1, the genomes lack an amidase to degrade the produced amides further to the corresponding carboxylic acid and ammonia, which could serve as an energy source. An example of a potential amidase is a putative formamidase identified in a clade B MAG obtained from the Rifle aquifer[52]. Other detoxification mechanisms include catalases and Fe or Mn superoxide dismutases (SOD) for reactive oxygen defense identified in both clade B genomes, and a periplasmic Cu-Zn SOD, which is present in La3-X1 and many other clade B genomes, but absent in La1-X1 (Fig. 3c, d).

An adaptation to oligotrophic conditions (Supplementary Table 1) might be the methionine salvage pathway (MSP) of La1-X1 (Fig. 3c). This pathway recycles the sulfur-containing intermediate 5′-methylthioadenosine back to methionine, thus allowing the use of reduced sulfur compounds under sulfur limitation. However, similar to the high-quality genome of the nitrite-oxidizing *Nitrospira lenta*[57] and other clade B genomes, the *mtnE* gene encoding the last step in this pathway is missing in La1-X1. Whether other enzymes might complete the MSP in these *Nitrospira* species remains to be determined.

## Nitrification and N₂O production activity

We have demonstrated that comammox *Nitrospira* potentially serves as crucial drivers of nitrification in coastal East Antarctica, and further investigated their survival strategies from a genomic perspective. Additionally, we confirmed the nitrification activity of comammox *Nitrospira* using DNA-SIP on two lake sediment samples (LA1 and LA2) and one soil sample (LS4). DNA-SIP incubations were conducted at 4 °C (for LA1) and 10 °C (for LA1, LA2, and LS4), and the production of $NO_3^-$ served as an indicator of nitrification activity (Fig. 4a and Supplementary Fig. 11). During the incubations, comammox *Nitrospira* were actively growing, as evidenced by the peak shifts of their DNA in the ¹³C-treatment (Fig. 4b and Supplementary Fig. 4). Subsequent sequencing of the *amoA* genes from the labeled DNA revealed clade B *amoA* OTUs, including those of comammox MAGs La1-X1 and La3-X1, in both the 4 °C and 10 °C incubations (Fig. 4c). Despite the temperature used in the DNA-SIP

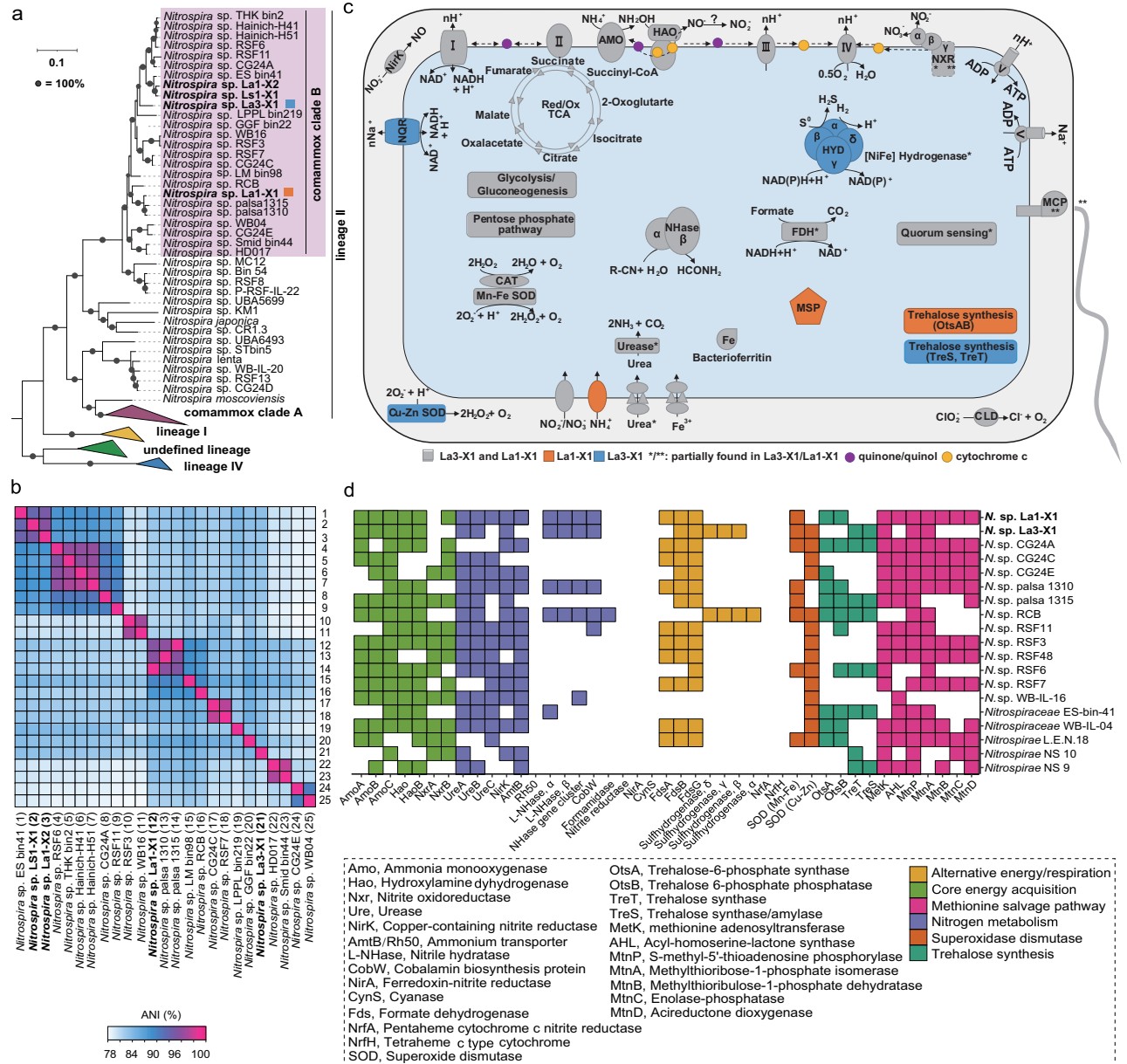

**Fig. 3 | Phylogenomic and metabolic analyses of the comammox *Nitrospira* metagenome-assembled genomes (MAGs), providing insights into their survival strategies in coastal East Antarctica. a** Maximum-likelihood phylogenomic tree of the genus *Nitrospira*, constructed using a concatenated alignment of 91 single-copy core genes. **b** Average nucleotide identity (ANI) analysis of selected clade B comammox *Nitrospira* genomes. **c** Comparative cellular metabolic diagram for MAGs *Nitrospira* sp. La1-X1 and *Nitrospira* sp. La3-X1. Selected features are depicted, including core pathways of chemolithoautotrophic ammonia and nitrite oxidation, alternative energy metabolism, and potential detoxification pathways. Colors denote the presence of specific genes in La1-X1 (orange), La3-X1 (blue), or both (gray). Asterisks represent incomplete features. Key enzymes and pathways are abbreviated as follows: AMO ammonia monooxygenase, CAT catalase, CLD chlorite dismutase, FDH formate dehydrogenase, HAO hydroxylamine dehydrogenase, HYD 3b [NiFe] hydrogenase, MCP methyl-accepting protein, MSP methionine salvage pathway, NHase nitrile hydratase, NirK Cu-dependent nitrite reductase, NQR Na$^+$-translocating NADH: ubiquinone oxidoreductase, NXR nitrite oxidoreductase, SOD superoxide dismutase. Enzyme complexes of the respiratory chains are labeled using Roman numerals. **d** Distribution of key metabolic features involved in N and alternative energy metabolism, the methionine salvage pathway, superoxide dismutase, and trehalose synthesis. In total, 19 *Nitrospira* genomes, including the two high-quality clade B comammox genomes (La1-X1 and La3-X1) obtained in this study, were analyzed. Features shown in white were not detected. Source data are provided as a Source Data file.

incubations not being identical to the in situ condition, our data strongly suggest an active role for clade B comammox *Nitrospira* in nitrification.

The addition of NH$_4^+$ resulted in an increase of NO$_3^-$ production, however, did not stimulate higher N$_2$O production compared to incubations without NH$_4^+$ addition (Fig. 4a), indicating the low N$_2$O production potential of comammox *Nitrospira*[55]. The addition of chlorate, a potential comammox-specific inhibitor[64], slightly reduced the production of NO$_3^-$ and N$_2$O (Fig. 4a). However, since chlorate can also influence other nitrite-oxidizing *Nitrospira* species as well as denitrifiers, the observed reductions in NO$_3^-$ and N$_2$O levels cannot be solely attributed to changes in comammox activity. Notably, prior research has shown the resilience of clade B comammox *Nitrospira* to freeze-thaw cycles, which underscores the endurance of this particular nitrifier group in environments with low and even subfreezing temperatures[65]. Raising the temperature (10 °C) stimulated the activity

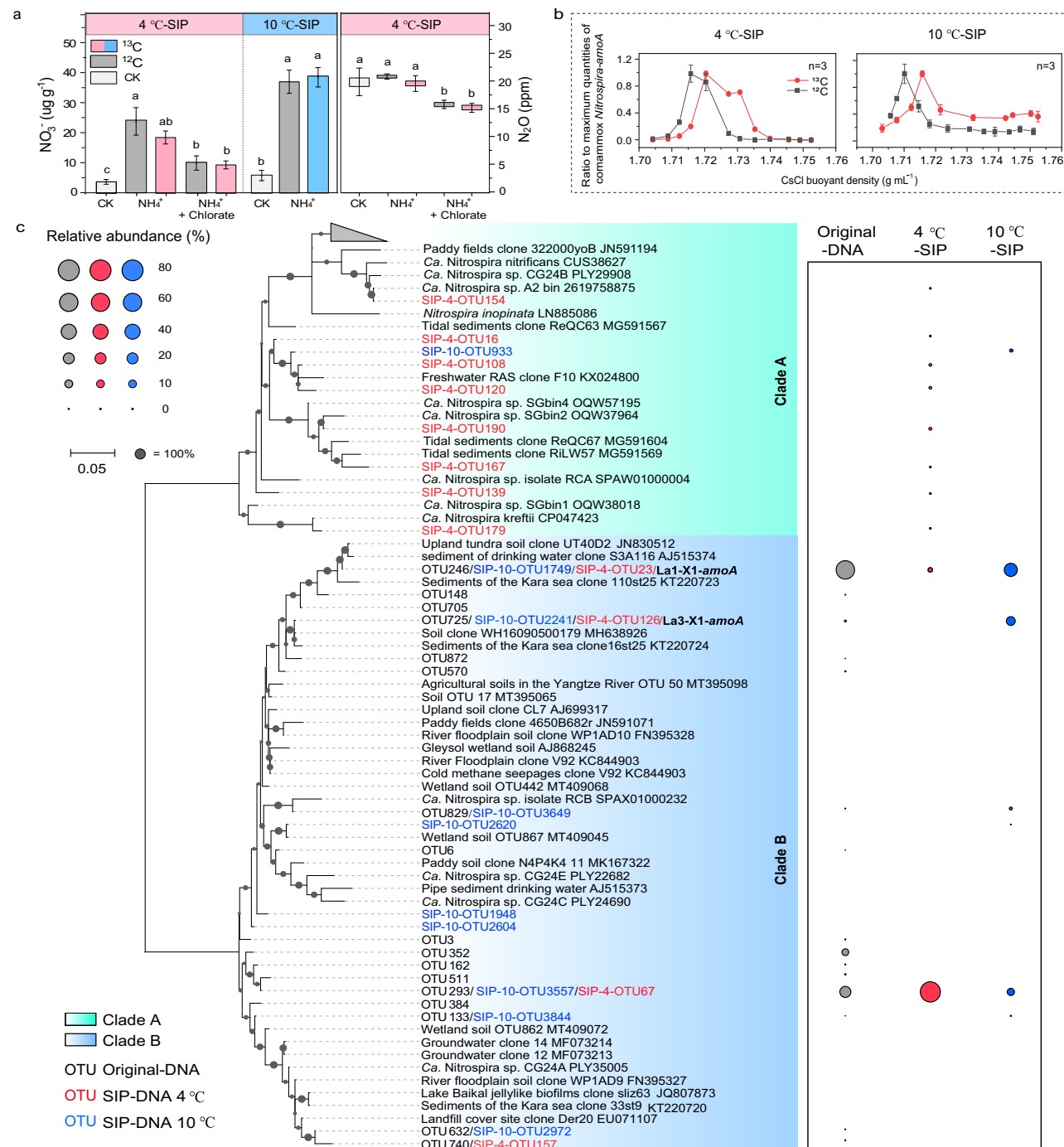

**Fig. 4 | The use of $^{13}$C-DNA stable isotope probing (SIP) to confirm the nitrification activity of clade B comammox _Nitrospira_ in LA1 lake sediment. a** The concentration of produced $NO_3^-$ (at 4 °C and 10 °C) and $N_2O$ (at 4 °C) after DNA-SIP incubations, using ammonium as substrate and chlorate as a specific inhibitor for comammox _Nitrospira_ (_t_-test, adjusted $P < 0.05$, $n = 3$ biological replicates). The incubation period for 10 and 4 °C incubations is 8 weeks (56 days) and 12 weeks (84 days), respectively. "CK" denotes samples that have not been amended with any substrate or inhibitor. **b** The quantitative distribution and relative abundance of comammox _Nitrospira amoA_ genes derived from DNA-SIP in $^{13}CO_2$ and $^{12}CO_2$-treated microcosms at 4 and 10 °C, respectively. Error bars represent standard errors ($n = 3$). **c** Maximum-likelihood phylogenetic tree of retrieved comammox _Nitrospira amoA_ gene sequences from the original sample DNA and $^{13}$C-DNA from 4 and 10 °C DNA-SIP incubation samples. The heatmap on the right displays the relative abundance of specific operational taxonomic units (OTUs) in different DNA samples. The additional DNA-SIP data pertaining to the 10 °C incubations of LA2 and LS4, as well as the _nxrB_ gene labeling data for LA1, are presented in the Supplementary Figs. 11–13.

of AOA (in LA2) and AOB (in LA2 and LS4) (Supplementary Fig. 4, Supplementary Results and Discussion), indicating that these nitrifiers are more competitive under elevated temperature conditions, an important finding to predict ecosystem changes in response to global warming.

## Implications and outlook

In this study, we elucidated the unique microbial N cycle in pristine and oligotrophic coastal Antarctic soil and lake sediments, identifying the microbial nitrification process as the primary pathway for $NO_3^-$ production. Our metagenomic and quantitative functional gene-targeted

analyses both revealed clade B comammox *Nitrospira* to be a key nitrifier in these environments. This finding not only expands our understanding of microbial diversity but also underscores the pivotal role of specific microbial groups in biogeochemical cycling in extreme environments. Our results also revealed fascinating patterns of niche differentiation between clade A and B comammox *Nitrospira* and canonical nitrifiers. Clade B species might have superior affinity for their substrate ammonia, and are potentially best adapted to survive and thrive in cold and oligotrophic environments. This niche differentiation could have significant implications for N-cycling in cold environments and it will be important to determine how the nitrifier and overall microbial communities will react to global warming.

In addition, we have successfully obtained in total of 1968 MAGs, significantly expanding the existing microbial genomic data for Antarctica. These MAGs provide invaluable insights into the microbial diversity and their metabolic capabilities in the extreme and unique environment of coastal Antarctica. The availability of these MAGs will undoubtedly facilitate further research into the intricate relationships between microbial communities, biogeochemical cycles, and climate change in polar regions and beyond.

This study underscores the importance of understanding the unique microbial N cycle in coastal Antarctica. We have uncovered clade B comammox *Nitrospira* to be a nitrification driver in low-temperature environments, and investigated their survival strategy and potential impact on climate change through the production of greenhouse gas $N_2O$[55,66]. It is crucial to further investigate how comammox *Nitrospira* evolve and survive in Antarctic ecosystems, meeting one of the six priorities for Antarctic science[67]. Recognizing the climatological significance of N-cycling and biogeochemical processes, future research should continue to monitor these microbes, as they could hold the key to understanding the broader implications of microbial activity on our planet's climate.

## Methods

### Sample collection and treatment

Samples were collected in Larsemann Hills (LH), the second-largest ice-free land in East Antarctica with an area of ~50 km². LH has a cold and dry continental climate, with an annual mean temperature of ~−10 °C and the temperatures occasionally above 0 °C in summer[68] lead to this region typically free of snow cover in summer. As a result of seasonal snow cover and the glacier melting, a number of land-locked lakes are developed in this region. Surface sediment (the upper ~5 cm) and surface water samples were collected from six lakes (LA1-LA6) in LHs in February 2020 (Fig. 1a). The surface sediment samples were collected using a stainless-steel spade on the shore of lakes, with typical water depths of ~150–200 cm. Approximately 1 L of surface water near the lake shore was also sampled with clean polyethylene (PE) bottles, a portion of which was used to measure temperature, pH, salinity, and conductivity utilizing a multi-probe water quality meter (YSI Professional Plus series; Supplementary Table 1). The remainder of the water samples was filtered using 0.22-μm polytetrafluoroethene (PTFE) filters for chemical analyses. Additionally, a surface soil sample (the top ~5 cm) was collected near each lake, ~100 m from the lake shore, using a clean stainless spatula. At each sampling site, larger gravels were removed firstly, and five soil sub-samples (four corners and the center of a square) were collected at a distance of 5–10 m and then mixed to obtain a representative sample (LS1-LS6). For a comprehensive understanding of N-cycling processes in the ice-free areas in coastal Antarctica, surface soil samples (SVL1−SVL10) were also collected in February 2022 at Inexpressible Island, South Victoria Land where the climate is comparable to that of LH, following the same sampling protocols. All of the sediment and soil samples were stored in sealed PE bags, and all samples were transported to the laboratory at temperatures of ~−20 °C for subsequent analysis.

### Physiochemical analysis

In the laboratory, ~50 g of the sediment and soil samples were freeze-dried in 50-mL clean centrifuge tubes (ALPHA 1-4/LD, Martin Christ Inc.). After drying, the samples were homogenized using an agate mortar and pestle, and subsequently passed through a 1 mm sieve for further chemical analysis. For determining total organic carbon (TOC) content, approximately 5 g of samples were digested with 10% HCl ($v/v$) to remove carbonate. Then, TOC was measured with an automatic element analyser (Elementar, VARIO EL III), with acetanilide used as the external standard. The detection limit (DL) of the TOC was estimated to be ~0.005%. All samples were measured in triplicate, yielding a relative standard deviation ($1\sigma$) of <10% for each sample. For chemical ion analysis, ~5 g of freeze-dried samples were placed in sterile 50-mL centrifuge tubes and suspended in 25 mL Milli-Q water (18.2 MΩ). The solution was then ultrasonicated for 40 min. The supernatant was first centrifuged for 15 min at 3000×$g$, and then filtered through 0.22-μm PTFE filters for nutrient determination. Nutrient concentrations ($NH_4^+$, $NO_3^-$, and $PO_4^{3-}$) in the sediment, soil extracts, and lake water samples were determined using an Aquion RFIC ion chromatograph (IC, Thermo Scientific, USA), equipped with the analytical columns CS12A (2 × 250 mm), AS11-HC (2 × 250 mm), methanesulfonic acid (MSA), and potassium hydroxide (KOH) as eluents for cations and anions, respectively. In addition, the concentrations of $SiO_3^{2-}$ were determined using an automated QuAAtro™ nutrient analyser (Seal Analytical Ltd., UK). During sample analysis, replicate determinations ($n = 5$) were performed, and $1\sigma$ for all species was <5%.

### Isotopic analysis of nitrate

The isotopic composition of $NO_3^-$ was determined using the bacterial denitrifier method at the Environmental Stable Isotope Laboratory of East China Normal University (ECNU-ESIL). Briefly, the denitrifying bacterium *Pseudomonas aureofaciens*, which lacks the $N_2O$ reductase enzyme, quantitatively transforms $NO_3^-$ into gaseous $N_2O$[69,70]. The $\delta^{15}N$ and $\delta^{18}O$ of the generated $N_2O$ were measured in duplicates using isotope-ratio mass spectrometry (IRMS, Thermo Scientific Delta V). The $\Delta^{17}O$ of $NO_3^-$ was separately analyzed through the thermal decomposition of $N_2O$ into $N_2$ and $O_2$[71], followed by measurements at m/z 32, 33, and 34 on the IRMS. The pooled standard deviation ($1\sigma_p$) was employed to ascertain the measurement precision of the overall denitrifier method[72,73]. The $1\sigma_p$ of all duplicate samples executed in at least two different batches was 0.6‰ for $\delta^{15}N$ ($n = 10$), 0.3‰ for $\delta^{18}O$ ($n = 10$), and 0.8‰ for $\Delta^{17}O$ ($n = 8$). However, due to the limited amounts of $NO_3^-$ in the samples, only three sediment and five surface soil samples were analyzed for $\Delta^{17}O$ of $NO_3^-$.

In addition, lake water stable isotopes ($\delta^{18}O$ and $\delta^2H$) were analyzed using laser absorption spectrometry (TIWA-45EP, Los Gatos Research, Inc.). To ensure quality control, replicate analyses ($n = 5$) were performed, yielding relative standard deviations of 0.05‰ and 0.2‰ for $\delta^{18}O$ and $\delta^2H$, respectively.

### DNA extraction, quantification, and sequencing analysis

Total DNA was extracted from 0.5 g sediment/soil samples using the Fast DNA SPIN kit (MP Biomedicals, Santa Ana, CA) according to the manufacturer's protocols. The final DNA quality and quantity were determined using Quant-iT PicoGreen dsDNA Assay Kit (Thermo Fisher Scientific, China). A detailed description of the quantitative PCR (qPCR) analysis for functional N-cycling genes is provided in the Supplementary Methods section. It also elaborates on the procedures for PCR and high-throughput amplicon sequencing, as well as the subsequent phylogenetic analysis of nitrification genes.

### Metagenomic sequencing

The total DNA from the original sediment and soil samples was sequenced on the Illumina HiSeq X ten platform using a 150-bp paired-end library at Beijing Novogene Biotech Co., Ltd. (Beijing, China).

The NEXTFLEX Rapid DNA-Seq Library Prep Kit 2.0 (Bioo Scientific, Austin, TX, USA) was used for DNA library preparation with an insert size of ~300 bp according to the manufacturer's recommendations. DNA was sheared using a Covaris S220 Focused Ultrasonicator to create 150 bp fragments. Subsequently, raw metagenomic sequencing data in FastQ format was generated using the RTA (Real-Time Analysis) v3.4.4 and bcl2fastq (btq) v2.16. Each individual sample yielded a range of 20–50 gigabases of sequencing data.

### Assembly and binning of metagenomes

Raw reads were processed using fastp v0.19.7[74] for adapter trimming, quality filtering, and per-read quality trimming. The fourteen quality-controlled metagenomes were individually assembled and co-assembled using MEGAHIT v1.1.3[75] with default parameters (k-mers: 21, 29, 39, 59, 79, 99, 119, 141). Each of the assembly was initially binned using the binning module (–metabat2 –maxbin2 –concoct; –metabat2 for co-assembly) in the metaWRAP pipeline[76] v1.3.2, and were consolidated using DAS Tool v1.1.2[77] with default parameters. After a dereplication check using dRep v3.0.0[78] (-comp 50 -con 10) and completeness and contamination evaluation using CheckM v1.1.3[79], 1968 MAGs were obtained, and the taxonomy of each MAG was assigned using GTDB-Tk v1.5.0[80] with the Genome Taxonomy Database (Release 06-RS202). In the process of annotating metabolic functions, the genomes that were extracted were examined using DIAMOND[81] v0.9.14 against 52 tailor-made protein databases, which consisted of marker genes representing common metabolic traits for energy conservation as well as C and N fixation[37]. To confirm the existence of crucial metabolic genes in the MAGs, maximum-likelihood phylogenetic trees were generated to confirm their phylogenetic affiliation The relative abundances (in percentage) for all dereplicated MAGs were determined by aligning each sample's clean paired-end reads to the MAGs utilizing CoverM v0.6.1 (https://github.com/wwood/CoverM) in genome mode, applying the default configurations. Moreover, the trimmed reads were incorporated into CLC Genomics Workbench version 20.0 (CLCBio, Qiagen, Germany), and the de novo assembly algorithm of CLC was employed to search for *amoA* and *nxrB* sequences. Contigs that contained either *amoA* or *nxrB* genes were selected for phylogenetic analysis, in conjunction with amplicon sequences.

### Phylogenomic analysis and genome annotation

Genomes and MAGs classified by the GTDB-Tk database R202[82] as *Nitrospiracea*, with estimated genome completeness ≥70% and contamination ≤10%, were downloaded from NCBI. Dereplication was performed using the drep v2.4.2[78] dereplicate workflow with cut-offs for estimated genome completion ≥70% and contamination ≤10%, but otherwise default settings. In addition to these 95 *Nitrospiracea* genomes, seven *Nitrospira* lineage IV and two clade B genomes were included in the phylogenetic analysis using the UBCG pipeline for the extraction and concatenated alignment of 91 single-copy core genes[83]. In addition, two *Leptospirillum* genomes (GCF_000284315.1, GCF_000299235.1) were included as outgroup. A maximum-likelihood phylogenetic tree was constructed using IQ-TREE v1.6.12[84] with 1000 ultrafast bootstrap replications and the GTR + F + I + G4 model identified by the implemented Modelfinder[85]. ANI analysis of all clade B genomes was performed using the OrthoANI[86]. Gene calling and automatic genome annotation were performed using the MicroScope platform[87], and annotations of selected features were manually checked and refined. For analyzing the distribution patterns of selected key features, 17 comammox clade B genomes were annotated using prokka v.1.14.6[88] with the "--gcode 11" and "--metagenome" options to obtain all protein sequences for generating a BLAST database. The distribution of selected key proteins within clade B was analyzed by conducting a BLASTp search against this database with default settings, except for an e-value cutoff of 1e-6. Only hits with an identity ≥35% (pident) and

a query coverage ≥80% (qcovs) were reported as present. Phylogenomic and ANI analyses were carried out on the retrieved AOA MAGs, using representative genomes (which include Group I.1a and Group I.1b-AOA) and Group I.1b *Nitrosocosmicus* AOA genomes as references, respectively.

### DNA-stable isotope probing (SIP) incubation and analysis

Lake sediments (LA1 and LA2) and the soil sample (LS4) were selected for DNA-SIP microcosm incubation experiments (Supplementary Fig. 10) to mimic a low-temperature oligotrophic environment. (Note that these three sampling locations are distributed almost uniformly throughout LH.) Microcosms were constructed in 120 mL serum bottles containing 10 g sediments or soil at 60% of maximum water-holding capacity and were incubated for 56 days at 10 °C in the dark. For each sample, two different treatments were established in triplicate microcosms. The $^{13}CO_2$ microcosms were amended with 5% (v/v) $^{13}CO_2$ (99 atom%; Sigma-Aldrich Co., St. Louis, MO, USA) plus approximately 5 μg $^{15}N$-$NH_4Cl$-N $g^{-1}$ dry weight soil/sediment (d.w.s.), while the $^{12}CO_2$ control treatments received 5% (v/v) $^{12}CO_2$ plus ~5 μg $^{14}N$-$NH_4Cl$-N $g^{-1}$ d.w.s. The water content was restored weekly over an 8-week incubation period by opening the bottles. Additional labeled or unlabeled $NH_4Cl$ was supplied (approximately once every 2–3 weeks) to maintain ~5 μg $NH_4Cl$-N $g^{-1}$ d.w.s. during the incubations. The sediment/soil samples were destructively sampled after 56 days of incubation and transferred immediately to –80 °C for subsequent molecular analysis. The remaining ~2 g of sediments were used for end-point quantification of $NH_4^+$, $NO_2^-$, and $NO_3^-$ concentrations. The LA1 sediment sample was additionally chosen for incubation at 4 °C, adhering to the same procedure previously outlined for a period of 84 days (12 weeks). In addition to the ammonium substrate, 50 μM chlorate, a specific comammox inhibitor[64], was introduced to study the activity of comammox *Nitrospira*. Moreover, the production of $N_2O$[64] was actively monitored during these 4 °C DNA-SIP incubations. The fractionation of DNA post-DNA-SIP incubations, along with the subsequent quantification analysis of functional nitrification groups, is described in the Supplementary Methods.

### Reporting summary

Further information on research design is available in the Nature Portfolio Reporting Summary linked to this article.

## Data availability

All AOA-, AOB-, and comammox *Nitrospira-amoA* and *Nitrospira-nxrB* OTU sequences obtained in this study were deposited in GenBank, with accession numbers MZ956347-MZ956585. Retrieved metagenome-assembled genomes (MAGs) have been uploaded to Figshare (https://doi.org/10.6084/m9.figshare.25435465). All raw sequencing reads of amplicon sequencing and metagenomes have been submitted to the National Center for Biotechnology Information (NCBI) under BioProject accession No. PRJNA855145. The database used in this study includes the GTDB database R06-RS202 (https://data.gtdb.ecogenomic.org/releases/release202/). The nitrifiers reference genomes used for comparative genomic analysis were downloaded from the NCBI Refseq database. Source data are provided with this paper.

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

## Acknowledgements

This work was supported by the National Natural Science Foundation of China (NSFC) (Nos. 42371064 to P.H., 42276243 to G.S., 42030411 to L.H., and 42230505 to M.L.), and the Program of Shanghai Academic Research Leader (Grant No. 20XD1421600 to G.S.), and by the Dutch

Research Council (NWO; Talent Program grants VI.Veni.192.086 and 016.Vidi.189.050 to S.L.). The LABGeM (CEA/Genoscope & CNRS UMR8030), the France Génomique, and the French Bioinformatics Institute national infrastructures (funded as part of Investissement d'Avenir program managed by Agence Nationale pour la Recherche, contracts ANR-10-INBS-09 and ANR-11-INBS-0013) are acknowledged for support within the MicroScope annotation platform.

## Author contributions

P.H., G.S. and M.L. conceived this study. P.H. and X.T. conducted the experimental work. P.H. and H.K. performed the comparative genomic analyses. X.T. analyzed the amplicon sequencing data. X.D. and P.H. analyzed the metagenomic data. D.W. and Z.L. collected the samples. Q.Z. did the physiochemical analyses. L.H. and S.L. contributed to the data interpretation. P.H. and G.S. wrote the manuscript, with contributions from all co-authors.

## Competing interests

The authors declare no competing interests.
