## [Peer Review File · Nature Communications]

Unveiling unique microbial nitrogen cycling and nitrification driver in coastal AntarcticaREVIEWER COMMENTS

Reviewer #1 (Remarks to the Author):

As an enthusiast of the microbial ecology of extreme environments, I consider this study quite exciting, refreshing and robust. I could be really picky about time-series measurements in different seasons, but I do understand the human, resource and weather limitations to develop these kinds of studies in such a harsh environment as Antarctica. So, I celebrate how the authors combined diverse methods to build a solid study to start filling the gap about the nitrification process and functional N-cycling genes in this unique environment. Besides the solid combination of culture dependent and independent work, I find the microcosm rate measurements at representative temperatures of the ecosystem a strong point of the study. The metagenomic analysis and the recovery of MAGs is robust and provides a great resource to the scientific community. The high point of the genomic analysis is the identification and predominance of commamox Nitrospira capable of performing the entire nitrification process. I found no methodological weaknesses in this manuscript and it is written clearly and concisely. I do have some minor comments that are not affecting the outcome of my decision, but I'd thank the authors for sending some feedback about them:

Results:

- Some sentences sound too general across the manuscript and can be more clear with the addition of some specific information in brackets. i.e. L. 209/210: ...suggesting their relatively high abundances (Supplementary Table 3). This could be improved with "...suggesting their relatively high abundances (>X%) (Supplementary Table 3)". This is an example but I've noticed the same pattern in different lines.
- Based on the high number of MAGs that the authors recovered, I'd love to see the estimations about how much of the microbial community is represented by this set of genomes/MAGs. SingleM appraise could help to get some estimations about this.

Methods:

- Not clear to me how the 52 metabolic marker genes database was built or if they used another similar database already reported. Include reference if necessary.

Figures:

- Fig. 1.a: Add distance bar for Larsemann/South Victoria maps.

Suppl. MyM

- There is a typo at the end of page 2. "in Larsemann Hills aera". Please correct it.
- OTUs vs ASVs (QIIME1 versus QIIME2): I've noticed that for the amoA amplicon sequencing, the authors used OTUs by clustering the sequences at 95% similarity cutoff. If they were presented only 16S data instead of amoA genes, I'd strongly recommend using QIIME2 and an ASVs approach instead OTUs/QIIME1 due to this approach being a little outdated and ASVs offer a higher resolution of the microbial community. So, my question out of curiosity is: why did the authors decide to use a clustering approach to define OTUs instead of using ASVs?

Reviewer #2 (Remarks to the Author):

In this manuscript, the authors delve into the microbial N cycling processes specific to the coastal regions of Antarctica, with a particular emphasis on East Antarctica. The work is a meticulously conducted analysis that paints a detailed picture of the N cycle in the region under investigation. The study is intriguing and represents significant work, encompassing physicochemical and isotopic analyses, quantification of functional genes, metagenomic and comparative genomic analyses, and ¹³C-DNA-based SIP experiments. The authors have observed unique N cycling processes and identified Commamox bacteria—previously not found—as the key players in nitrification within Antarctica's oligotrophic and ice-free areas. This discovery is poised to captivate those researching biogeochemical N cycling and polar science.

From a methodological standpoint, the authors have employed a comprehensive array of tools, including isotopic analysis of nitrate, extensive quantification of functional N-cycling marker genes, deep metagenomic sequencing, and detailed comparative genomic analysis. Obtaining a multitude of MAGs, especially some high-quality Commamox genomes, is not trivial. Furthermore, implementing the ¹³C-DNA-based SIP method at two different temperatures (4 and 10 °C) on selected sediment and soil samples validated the role of detected Commamox Nitrospira in nitrification. The authors have utilized virtually available methods to substantiate their findings.

The manuscript is well-structured and written, making for an engaging read. Despite the complexity and breadth of content related to microbial N cycling quantification, the authors have presented their findings in a clear, structured, and succinct manner. The logical progression of the narrative effectively draws the reader's attention to the novel nitrification driver, clade B Commamox Nitrospira. Since their discovery, this is a significant stride, marking the first in-depth investigation of complete nitrifiers in the Antarctic environment. The findings that Commamox bacteria, particularly clade B, are highly abundant and active in the area studied are surprising. They will inspire further research into these microbial communities' ecophysiological roles and evolutionary trajectories in cold and oligotrophic environments. Overall, this work meets the journal's requirements and is of high value to the reader. And I have some comments and suggestions before being accepted.

Major concerns:

- (1) The sampling time is precise. However, we don't know when the incubation experiments were conducted, which was related to nitrogen transformation and microbial activities.
- (2) The relative abundance of microbial N cycle genes is defined by copies of ng DNA in Figure 1bcd. Some works indicated that 'Measuring the total count and concentration of microbes or genes (absolute abundance) provides a richer source of information than relative abundance and can correct some conclusions drawn from relative abundance data (<https://doi.org/10.1038/s41587-023-01754-3>)'. Thus, it would be good to have absolute quantitative data if possible.
- (3) De-noising algorithm analysis (ASVs), rather than OTUs, is more accurate and recommended.
- (4) Why were only three soil (LS4) and sedimental (LA1 and LA2) samples selected for the SIP experiment? What were the selection criteria?

(5) The ammonium and nitrate contents were low (Supplementary Table 1). During the SIP incubation, approximately 5 $\mu\text{g g}^{-1}$ of ^{15}N labeled NH_4^+ was added as the substrate. About 40 $\mu\text{g g}^{-1}$ of nitrate and ~ 20 ppm of N_2O were detected after the 56-day incubation (Figure 4a). Do you have the isotopic abundance values of nitrate? This is important to illustrate the nitrogen transformation processes.

Minor concerns:

- (1) The abstract could be more succinct. Some abbreviations can be omitted.
- (2) Did the authors find any exciting properties in the AOA genomes? Especially in the nutrient-poor environment of this study.
- (3) Line 48: Consider referring to it as the 'ice-free Antarctic region,' 'Coastal East Antarctica,' or 'Coastal Antarctica'.
- (4) Lines 63-69: lack references.
- (5) Line 244: 'La3-bin1' should be corrected to 'La3-X1'. Right?
- (6) Line 287: Please correct the spacing issue in 'Fe or Mn'.
- (7) Line 437: Number 'seven' should be replaced with 'fourteen'. Right?
- (8) Line 458: Please confirm the term 'all available' as databases are regularly updated.
- (9) Lines 485-486: Could you clarify what 'in situ flooding conditions' mean?
- (10) Line 504: It should read 'OTU sequences'.
- (11) Fig. 3a: Please double-check the phylogenomic tree; 'palsa 1315' appears twice.
- (12) Fig. 3d: AHL is missing from the bottom part of the figure.
- (13) Figure 4a: What is 'CK' treatment? No description is found in Section 'Material and Methods' either.
- (14) Supplementary Fig. 6 and Supplementary Fig. 7: Please confirm the accuracy of the colored shadows.
- (15) Supplementary Fig. 9: The stray '25' is in the middle of the figure; please remove it.

We are grateful to the reviewers for their careful reading of the manuscript, constructive criticisms, positive comments and encouraging remarks, all of which have significantly enhanced the quality of our resubmitted paper. Below, we have addressed each reviewer's comments (noted in black) with our responses (in blue):

Reviewers' Comments

Reviewer #1 (Remarks to the Author):

As an enthusiast of the microbial ecology of extreme environments, I consider this study quite exciting, refreshing and robust. I could be really picky about time-series measurements in different seasons, but I do understand the human, resource and weather limitations to develop these kinds of studies in such a harsh environment as Antarctica. So, I celebrate how the authors combined diverse methods to build a solid study to start filling the gap about the nitrification process and functional N-cycling genes in this unique environment. Besides the solid combination of culture dependent and independent work, I find the microcosm rate measurements at representative temperatures of the ecosystem a strong point of the study. The metagenomic analysis and the recovery of MAGs is robust and provides a great resource to the scientific community. The high point of the genomic analysis is the identification and predominance of commamox *Nitrospira* capable of performing the entire nitrification process. I found no methodological weaknesses in this manuscript and it is written clearly and concisely. I do have some minor comments that are not affecting the outcome of my decision, but I'd thank the authors for sending some feedback about them:

Response: We thank the reviewer for these very positive comments on the general significance of our manuscript.

Results:

- Some sentences sound too general across the manuscript and can be more clear with the addition of some specific information in brackets. i.e. L. 209/210: ...suggesting their relatively high abundances (Supplementary Table 3). This could be improved with "...suggesting their relatively high abundances (>X%) (Supplementary Table 3)". This is an example but I've noticed the same pattern in different lines.

Response: We thank the reviewer for this point. We have revised the whole manuscript accordingly (lines 120, 125-130 and 165).

- Based on the high number of MAGs that the authors recovered, I'd love to see the estimations about how much of the microbial community is represented by this set of genomes/MAGs. SingleM appraise could help to get some estimations about this.

Response: We appreciate the reviewer's comment. We used CoverM (version 0.6.1) to estimate the coverage of the MAGs in the samples investigated. The total relative abundances ranged from 29.6% to 54.5%, with an average of 41.6% (see Supplementary Data 2).

Methods:

- Not clear to me how the 52 metabolic marker genes database was built or if they used another similar database already reported. Include reference if necessary.

Response: We thank the reviewer for this point. The metabolic marker genes database is based on a previous publication. The related reference has been added accordingly (lines 161 and 446).

Figures:

-Fig. 1.a: Add distance bar for Larsemann/South Victoria maps.

Response: We thank the reviewer for this point. Distance bars were added to the maps. Additionally, we revised Fig. 1 in accordance with the reviewers' comments.

Suppl. MyM

- There is a typo at the end of page 2. "in Larsemann Hills aera". Please correct it.

Response: We thank the reviewer for this point. It has been corrected.

- OTUs vs ASVs (QIIME1 versus QIIME2): I've noticed that for the amoA amplicon sequencing, the authors used OTUs by clustering the sequences at 95% similarity cutoff. If they were presented only 16S data instead of amoA genes, I'd strongly recommend using QIIME2 and an ASVs approach instead OTUs/QIIME1 due to this approach being a little outdated and ASVs offer a higher resolution of the microbial community. So, my question out of curiosity is: why did the authors decide to use a clustering approach to define OTUs instead of using ASVs?

Response: We appreciate the reviewer's point. We indeed used the QIIME2 platform, not QIIME1, for the analysis. We have updated the related information in the Supplementary Methods. The QIIME2 codes used in this study are also provided in Appendixes in the Supplementary Information.

We agree with the reviewer that for a 16S rRNA gene dataset, ASV analysis is the most commonly used and recommended method. For functional marker genes, such as *nxB*, clustering based on 95% similarity is recommended (Pester et al., 2014). This is because multiple non-identical copies of the marker gene within one genome can lead to an overestimation of diversity. Therefore, we used a more stringent approach and clustered the ASV sequences obtained via QIIME2 into OTUs using a 95% similarity threshold.

Reviewer #2 (Remarks to the Author):

In this manuscript, the authors delve into the microbial N cycling processes specific to the coastal regions of Antarctica, with a particular emphasis on East Antarctica. The work is a meticulously conducted analysis that paints a detailed picture of the N cycle in the region under investigation. The study is intriguing and represents significant work, encompassing physicochemical and isotopic analyses, quantification of functional genes, metagenomic and comparative genomic analyses, and ^{13}C -DNA-based SIP experiments. The authors have observed unique N cycling processes and identified Commamox bacteria—previously not found—as the key players in nitrification within Antarctica's oligotrophic and ice-free areas. This discovery is poised to captivate those researching biogeochemical N cycling and polar science.

From a methodological standpoint, the authors have employed a comprehensive array of tools, including isotopic analysis of nitrate, extensive quantification of functional N-cycling marker genes, deep metagenomic sequencing, and detailed comparative genomic analysis. Obtaining a multitude of MAGs, especially some high-quality Commamox genomes, is not trivial. Furthermore, implementing the ^{13}C -DNA-based SIP method at two different temperatures (4 and 10 °C) on selected sediment and soil samples validated the role of detected Commamox Nitrospira in nitrification. The authors have utilized virtually available methods to substantiate their findings.

The manuscript is well-structured and written, making for an engaging read. Despite the complexity and breadth of content related to microbial N cycling quantification, the authors have presented their findings in a clear, structured, and succinct manner. The logical progression of the narrative effectively draws the reader's attention to the novel nitrification driver, clade B Commamox Nitrospira. Since their discovery, this is a significant stride, marking the first in-depth investigation of complete nitrifiers in the Antarctic environment. The findings that Commamox bacteria, particularly clade B, are highly abundant and active in the area studied are surprising. They will inspire further research into these microbial communities' ecophysiological roles and evolutionary trajectories in cold and oligotrophic environments.

Overall, this work meets the journal's requirements and is of high value to the reader. And I have some comments and suggestions before being accepted.

Response: We thank the reviewer for these very positive comments on the general significance of our work and on the complementary nature of the methodology used to reveal novel insights into the N-cycling processes in the Antarctic ecosystem.

Major concerns:

- (1) The sampling time is precise. However, we don't know when the incubation experiments were conducted, which was related to nitrogen transformation and microbial activities.

Response: We appreciate the reviewer's comment. The soil and lake sediment samples were immediately stored at -20 °C after collection and were maintained at this temperature during transportation. The samples arrived at the low temperature laboratory (-20°C) in East China Normal University in May 2020. The 10 °C -SIP incubation was performed in June 2020. The 4°C SIP incubation was conducted following the 10 °C SIP incubation.

We agree with the reviewer that the incubation time may affect nitrogen transformation and microbial activities. However, considering the long-term low temperature in the Antarctic region, storing the samples at -20°C may have little effect on the activity of cryogenic microorganisms. Additionally, the successful labeling of functional nitrifiers and the observed nitrate production activity further support the recovery of related microbes.

(2) The relative abundance of microbial N cycle genes is defined by copies of ng DNA in Figure 1bcd. Some works indicated that 'Measuring the total count and concentration of microbes or genes (absolute abundance) provides a richer source of information than relative abundance and can correct some conclusions drawn from relative abundance data (<https://doi.org/10.1038/s41587-023-01754-3>)'. Thus, it would be good to have absolute quantitative data if possible.

Response: We appreciate the reviewer's comment. We agree with the reviewer and have provided the quantitative source data in both copies. g⁻¹ and copies. ng⁻¹ DNA formats in Supplementary Data 1 and revised Fig. 1 in accordance with the reviewers' comments. The reason we used copies. ng⁻¹ DNA data in Fig.1, rather than copies. g⁻¹, is due to the high heterogeneity of the soil/sediment samples (mixing with sands or even rocks). These samples have very low organic matter/biomass and are resistant to extraction of high-quantity/quality DNA, making copies. ng⁻¹ DNA data more meaningful for comparisons. This is also why we did not perform a metagenomic analysis on all soil samples (2 out of 10 sequenced) from South Victoria Land, due to the limitations posed by the extraction of high-quantity/quality DNA.

(3) De-noising algorithm analysis (ASVs), rather than OTUs, is more accurate and recommended.

Response: We appreciate the reviewer's comment, which was also raised by Reviewer 1 (see comment and response above). All our amplicon data were analyzed through QIIME2 ASVs, with representative sequences clustered based on >95% similarity being named OTUs. While ASV analysis is very common for the 16S rRNA gene, there was a concern about the potential for overestimating functional marker gene diversity due to non-identical gene copies within the same genome. As a result, we adopted a more stringent approach and clustered the functional ASVs into OTUs.

(4) Why were only three soil (LS4) and sedimental (LA1 and LA2) samples selected for the SIP experiment? What were the selection criteria?

Response: We appreciate the reviewer's point. In fact, the SIP experiment is not a high-throughput method and it's not realistic to apply it to all samples. Furthermore, there is no significant difference among the samples in terms of the abundances of functional n-cycling genes (Fig. 1), as well as the metagenomic output (Fig. 2). Therefore, it's not necessary to conduct SIP incubations with all samples. The reasons we chose LA1, LA2, and LS4 for further SIP analysis are: 1) these samples represent different types (soil and sediment) and 2) they are distributed almost uniformly throughout the study area LH; that is, LA1 is next to the Antarctic ice sheet; LA2 is in the center of LH; and LS4 is adjacent to the coastal line (Fig.1). This point was included in the revised manuscript (lines 485-486).

(5) The ammonium and nitrate contents were low (Supplementary Table 1). During the SIP incubation, approximately 5 $\mu\text{g g}^{-1}$ of ^{15}N labeled NH_4^+ was added as the substrate. About 40 $\mu\text{g g}^{-1}$ of nitrate and ~ 20 ppm of N_2O were detected after the 56-day incubation (Figure 4a). Do you have the isotopic abundance values of nitrate? This is important to illustrate the nitrogen transformation processes.

Response: We appreciate the reviewer's comment. For the 10 °C -SIP, the incubation period was 8 weeks, during which the NH_4^+ substrate was added weekly in approximately 5 ppm increments, resulting in a final cumulative product of around 40 $\mu\text{g g}^{-1}$ nitrate. For the 4°C -SIP, the incubation period was 12 weeks, and the NH_4^+ substrate was added every 2-3 weeks. This also resulted in accumulation of nitrate. Given the extremely low initial quantities of NH_4^+ and NO_3^- in the original samples (as shown in Supplementary Table 1), all accumulated NO_3^- can be considered as a product derived from the NH_4^+ substrate.

We agree with the reviewer that the ratio between labeled and unlabeled nitrate is an intriguing aspect to consider when understanding the significance of internal N sources versus added NH_4^+ during dissimilatory processes. However, the primary aim of the experiment was to label metabolically active cells that utilized the provided labeled N and C for assimilatory processes, incorporating these elements into their DNA. The reason we applied ^{15}N labeled NH_4^+ , in addition to ^{13}C - CO_2 during the SIP incubation, was to enhance the incorporation of heavy C/N into active nitrifiers. These methods are commonly employed in similar SIP incubation setups (Sun et al., 2022; Wang et al., 2019).

Minor concerns:

(1) The abstract could be more succinct. Some abbreviations can be omitted.

Response: We thank the reviewer for this comment. We have removed the abbreviations and shortened the abstract within 150 words, as required by *Nature Communications*.

(2) Did the authors find any exciting properties in the AOA genomes? Especially in the nutrient-poor environment of this study.

Response: We appreciate the reviewer's point. AOA has previously been reported in coastal Antarctic soils (Ortiz et al., 2020) and thus is not the focus of this study. Nevertheless, we screened the annotations of the retrieved AOA genomes and did not find uniquely significant features so far. As more Antarctic-originated AOA genomes or isolates become available, we will be able to conduct further genomic or even physiological investigations regarding their survival strategies and metabolic potentials.

Line 48: Consider referring to it as the 'ice-free Antarctic region,' 'Coastal East Antarctica,' or 'Coastal Antarctica'.

Response: We thank the reviewer for spotting this. We have made revisions accordingly.

(3) Lines 63-69: lack references.

Response: We thank the reviewer for this point. Related references have been cited (lines 56-61).

(4) Line 244: 'La3-bin1' should be corrected to 'La3-X1'. Right?

Response: We thank the reviewer for this correction. Changes have been made accordingly (lines 230-241).

(5) Line 287: Please correct the spacing issue in 'Fe or Mn'.

Response: We thank the reviewer for this correction. Change has been made accordingly (line 283).

(6) Line 437: Number 'seven' should be replaced with 'fourteen'. Right?

Response: We thank the reviewer for this correction. We have changed the number accordingly (line 435).

(7) Line 458: Please confirm the term 'all available' as databases are regularly updated.

Response: We thank the reviewer for this point. Changes have been made accordingly (line 457).

Lines 485-486: Could you clarify what 'in situ flooding conditions' mean?

Response: We thank the reviewer for this point. Changes have been made accordingly (lines 484-485).

(8) Line 504: It should read 'OTU sequences'.

Response: We thank the reviewer for this point. Changes have been made as suggested (lines 506-507).

(9) Fig. 3a: Please double-check the phylogenomic tree; 'palsa 1315' appears twice.

Response: We thank the reviewer for this correction. The correction has been made (Fig.3). In addition, we double-checked all our figures to ensure there are no similar mistakes. The information of reference genomes, including those for comammox *Nitrospira*, AOA and NOB *Nitrospira*, is listed in Supplementary Table 4. Moreover, the source data for comammox and AOA ANI analysis can be found in Supplementary Data 5 and Supplementary Data 6, respectively.

(10) Fig. 3d: AHL is missing from the bottom part of the figure.

Response: We thank the reviewer for this point. We have added the related information of AHL accordingly (Fig.3).

(11) Figure 4a: What is 'CK' treatment? No description is found in Section 'Material and Methods' either.

Response: We appreciate the reviewer's suggestion. We have accordingly added the related information for 'CK' in the legend of Fig. 1.

(12) Supplementary Fig. 6 and Supplementary Fig. 7: Please confirm the accuracy of the colored shadows.

Response: We thank the reviewer for this point. Changes have been made accordingly.

(13) Supplementary Fig. 9: The stray '25' is in the middle of the figure; please remove it.

Response: We thank the reviewer for this point. Changes have been made accordingly.

References:

- Ortiz M, Bosch J, Coclet C, Johnson J, Lebre P, Salawu-Rotimi A, et al. Microbial nitrogen cycling in Antarctic soils. *Microorganisms* 2020; 8.
- Pester M, Maixner F, Berry D, Rattei T, Koch H, Lucker S, et al. NxrB encoding the beta subunit of nitrite oxidoreductase as functional and phylogenetic marker for nitrite-oxidizing *Nitrospira*. *Environ. Microbiol.* 2014; 16: 3055-71.
- Sun X, Zhao J, Zhou X, Bei Q, Xia W, Zhao B, et al. Salt tolerance-based niche differentiation of soil ammonia oxidizers. *ISME J.* 2022; 16: 412-422.
- Wang B, Qin W, Ren Y, Zhou X, Jung MY, Han P, et al. Expansion of Thaumarchaeota habitat range is correlated with horizontal transfer of ATPase operons. *ISME J.* 2019; 13: 3067-3079.

REVIEWERS' COMMENTS

Reviewer #1 (Remarks to the Author):

Thank you to the authors for addressing all my comments and making the changes accordingly. No major comments or changes required from my end.

Just a small clarification about CoverM (approach used by the authors) and SingleM appraise (approach suggested by me in the previous revision). Please, keep this in mind for your future work.

Both approaches complement each other, but they indicate/measure different things. While CoverM is giving you the relative abundances of the MAGs in the community, SingleM is estimating the total number of taxa in the community based on binned and unbinned sequences and indicating how many of that total are represented by the recovered MAGs.

i.e. CoverM output

MAG_1 5%

MAG_2 3%

MAG_3 2%

Subtotal: 10% of mapped reads

i.e. SingleM output:

10 different species detected by SingleM appraise and present in your community, but only 3 species represented in your MAGs in the above example. 3 out of 10 (30% of the total species recovered in the MAGs).

I hope this clarification helps. Take a look at this link for your future work.

<https://bio-protocol.org/exchange/minidetail?type=30&id=9291876>

Reviewer #2 (Remarks to the Author):

I am glad that the authors have revised the manuscript according to the comments, and this manuscript is much improved in terms of data presentation and description. I consider that this manuscript has fulfilled the requirements for publication in Nature Communication.

Some minor suggestions:

(1) The annual mean air temperature is \sim -10 °C in the sampling sites. I agree that storing the samples at -20 °C may have little effect on the microbial activities. What's the in-situ temperature of soil and sediment in this region? This data will better indicate the low impact of sample preservation temperature on microbial activities.

(2) Compared to the in situ condition, the incubation condition of ¹³C-DNA-SIP was more specific. Thus, it is recommended that the limitations of the SIP results in this manuscript should be stated.

We thank the reviewers for their encouraging remarks and further comments. The manuscript has been revised accordingly. Below, we have addressed each reviewer's comments (noted in black) with our responses (in blue):

REVIEWERS' COMMENTS

Reviewer #1 (Remarks to the Author):

Thank you to the authors for addressing all my comments and making the changes accordingly.

No major comments or changes required from my end.

Response: We are delighted to receive this news and wish to express our gratitude to the reviewer for the insightful comments on our manuscript.

Just a small clarification about CoverM (approach used by the authors) and SingleM appraise (approach suggested by me in the previous revision). Please, keep this in mind for your future work.

Both approaches complement each other, but they indicate/measure different things. While CoverM is giving you the relative abundances of the MAGs in the community, SingleM is estimating the total number of taxa in the community based on binned and unbinned sequences and indicating how many of that total are represented by the recovered MAGs.

i.e. CoverM output

MAG_1 5%

MAG_2 3%

MAG_3 2%

Subtotal: 10% of mapped reads

i.e. SingleM output:

10 different species detected by SingleM appraise and present in your community, but only 3 species represented in your MAGs in the above example. 3 out of 10 (30% of the total species recovered in the MAGs).

I hope this clarification helps. Take a look at this link for your future work.

<https://bio-protocol.org/exchange/minidetail?type=30&id=9291876>

Response: We appreciate the reviewer's insightful comments on the distinct algorithms and outputs between CoverM and SingleM. Indeed, we plan to incorporate SingleM evaluations in our future work.

Reviewer #2 (Remarks to the Author):

I am glad that the authors have revised the manuscript according to the comments, and this manuscript is much improved in terms of data presentation and description. I consider that this manuscript has fulfilled the requirements for publication in Nature Communication.

Response: We are delighted to receive this news, and we express our gratitude for the reviewer's constructive feedback and corrections.

Some minor suggestions:

(1) The annual mean air temperature is ~ -10 °C in the sampling sites. I agree that storing the samples at -20 °C may have little effect on the microbial activities. What's the in-situ temperature of soil and sediment in this region? This data will better indicate the low impact of sample preservation temperature on microbial activities.

Response: We appreciate the reviewer for raising this point. The in-situ temperatures of soil and sediment in this region while we were collecting the samples were also low

(around 0 °C), even in summer. This also suggests that the preservation of samples at low temperatures has a minimal impact on microbial activities.

(2) Compared to the in situ condition, the incubation condition of ^{13}C -DNA-SIP was more specific. Thus, it is recommended that the limitations of the SIP results in this manuscript should be stated.

Response: We appreciate the reviewer for this comment. The limitations of the SIP results in this manuscript have already been stated in the manuscript (Lines 309-310).